# Protocol for a feasibility study of OnTrack: a digital system for upper limb rehabilitation after stroke

Gianpaolo Fusari [1,2] Ella Gibbs,[2] Lily Hoskin,[2] Daniel Dickens,[2] Melanie Leis,[3] Elizabeth Taylor [4] Fiona Jones,[4] Ara Darzi[2]

¹Helix Centre, Royal College of Art, London, UK
²Helix Centre, Imperial College London, London, UK
³Big Data and Analytical Unit, Institute of Global Health Innovation, Imperial College London, London, UK
⁴Faculty of Health Social Care and Education, Kingston and St George's University of London, London, UK

**Correspondence to**
Gianpaolo Fusari;
gianpaolo@helixcentre.com

## ABSTRACT

**Introduction** Arm weakness is a common problem after stroke (affecting 450 000 people in the UK) leading to loss of independence. Repetitive activity is critical for recovery but research shows people struggle with knowing what or how much to do, and keeping track of progress. Working with more than 100 therapists (occupational therapists and physiotherapists) and patients with stroke, we codeveloped the OnTrack intervention—consisting of software for smart devices and coaching support—that has the potential to address this problem. This is a protocol to assess the feasibility of OnTrack for evaluation in a randomised control trial.

**Methods and analysis** A mixed-method, single-arm study design will be used to evaluate the feasibility of OnTrack for hospital and community use. A minimum sample of 12 participants from a stroke unit will be involved in the study for 14 weeks. During week 1, 8 and 14 participants will complete assessments relating to their arm function, arm impairment and activation. During weeks 2–13, participants will use OnTrack to track their arm movement in real time, receive motivational messages and face-to-face sessions to address problems, gain feedback on activity and receive self-management skills coaching. All equipment will be loaned to study participants. A parallel process evaluation will be conducted to assess the intervention's fidelity, dose and reach, using a mixed-method approach. A public and patient involvement group will oversee the study and help with interpretation and dissemination of qualitative and quantitative data findings.

**Ethics and dissemination** Ethical approval granted by the National Health Service Health Research Authority, Health and Care Research Wales, and the London—Surrey Research Ethics Committee (ref. 19/LO/0881). Trial results will be submitted for publication in peer review journals, presented at international conferences and disseminated among stroke communities. The results of this trial will inform development of a definitive trial.

**Trial registration number** NCT03944486.

### Strengths and limitations of this study

► This is a feasibility trial of a novel intervention which employs an integrated approach for tracking arm activity and coaching with the aim of increasing stroke survivors' confidence and ability to use their impaired arm in daily activities, increasing the opportunities for repetitive rehabilitation (repeating a movement or series of movements with a rehabilitative or functional aim).

► Patient and public involvement (PPI) with more than 100 stroke survivors, carers and clinicians have contributed to our needs-finding phase, codesigned OnTrack and informed the feasibility study. A new PPI group will oversee the running of the study and help with interpretation of qualitative and quantitative data findings.

► An independent process evaluation will provide detailed information about implementation, context and the mechanisms of impact of the intervention. Findings will help in the understanding of intervention fidelity and training needs required for a definitive trial.

► For pragmatic reasons, the study uses a non-randomised design carried out at a single site—this will limit understanding about randomisation and recruitment.

► Participants will not be followed up after intervention period; however, participant views will be sought regarding appropriate follow-up times in a subsequent definitive trial.

who live in the country have some form of disability, significantly contributing to the loss of independence and feeling of isolation that they experience.[2 3] Furthermore, stroke is estimated to cost UK society £26 billion every year, with the vast majority of these costs borne by the informal care sector.[2]

Upper limb (arm) weakness is the main cause of physical impairment affecting 75% of disabled stroke survivors; this equates to around 450 000 people in the UK.[2] Dose-intensive repetitive rehabilitation is widely accepted as the 'gold-standard' for regaining ability after stroke; however, National Health

## INTRODUCTION

Every year around the world over 15 million people experience a stroke, leaving 5 million people with a permanent disability.[1] Stroke is the leading cause of disability in the UK; half of the nearly 1.2 million stroke survivors

Service (NHS) resources are often limited and unable to provide this.[4] A recent Cochrane review of over 500 trials failed to yield high-quality practice recommendations for interventions for the upper limb.[5] Arm recovery after stroke is a national research priority.[6] There is a correlation between physical activity after stroke and the ability to perform activities of daily living (most of which involve the use of the arm).[7] Despite this evidence, studies suggest that the actual time patients are active is minimal.[8] Many current approaches to increasing repetitive rehabilitation focus on improving the prescribed rehabilitation sessions (typically lasting 45–60 min), often by employing gamification techniques.[9 10]

While this is important, there is untapped potential to increase repetitive rehabilitation by targeting the large proportion of the day where patients are going about their daily activities and can use their arm movement (however small) to a greater extent. Capacity for activity could be increased further by using self-management methods as demonstrated by several different programmes in stroke and other long-term conditions.[11–14] This has informed the development of OnTrack which aims to increase opportunities for activity by improving individuals' self-management skills through tailored support and real-time activity feedback on their arm movement.

An unpublished ethnographic study conducted by the Helix Centre (funded by Innovate UK) confirmed what other studies have shown[7 8 15] that patients struggle to see and keep track of improvements; this impacts their motivation and leaves them dependent on therapists for feedback. Stroke survivors often report feeling unsupported after leaving hospital and not knowing how to best help themselves improve their arm function.[16–18] Feedback gathered from over 100 stroke survivors and clinicians was the basis for developing the OnTrack intervention.

A proof-of-concept test of OnTrack gathered data from a small group of patients (n=7) and confirmed that the intervention was safe and generally users could understand how and when to use it. Participants reported that they were more aware of their impaired arm and had increased confidence in using it for new tasks. A 20% mean increase in minutes of activity on the impaired arm was observed. The work conducted to date is unpublished and has some limitations; however, it has shaped the intervention and suggests that OnTrack has the potential to be a scalable solution that requires minimal training and could be used in conjunction with NHS services to help increase the overall amount of activity performed with the impaired arm. This study will assess the feasibility of the OnTrack intervention and inform the design of a definitive randomised controlled trial (RCT) to evaluate its clinical effectiveness, and follows the Standard Protocol Items: Recommendations for Interventional Trials guidelines.[19]

## METHODS AND ANALYSIS
### Aims and objectives

The primary aim is to evaluate the feasibility of an RCT to test the effectiveness of the OnTrack intervention for upper limb rehabilitation after stroke.

The objectives are to:
► Assess the feasibility of recruitment from hyperacute and acute stroke units, and rehabilitation wards to ascertain strategy and recruitment rates.
► Assess dropout rates by observing adherence and compliance with the intervention.
► Understand the acceptability and usability of the intervention by stroke survivors.
► Understand the acceptability of study procedures by healthcare professionals.
► Explore implementation fidelity, dose and reach of the OnTrack intervention.

The study will also collect clinical outcomes regarding arm function, impairment and activation to identify an appropriate primary outcome, and to estimate parameters for a sample size calculation for an RCT (table 1).

| Table 1 | Outcome measures | |
|---|---|---|
| **Concept** | **Assessment** | **Week of administration** |
| Patient activation/engagement | Patient Activation Measure | 1, 8, 14 |
| Arm impairment | Fugl-Meyer Assessment for Upper Extremity | 1, 8, 14 |
| Arm function | Upper Extremity Motor Activity Log-14 | 1, 8, 14 |
| Gross level of disability | modified Rankin Scale | 1, 8, 14 |
| Arm pain | Visual Analogue Scale | 1, 8, 14 |
| Cognitive impairment | Montreal Cognitive Assessment | 1, 8, 14 |
| Arm neglect | Albert's Test | 1, 8, 14 |
| Quality of life | EQ-5D-5L | 1, 8, 14 |
| Arm function | Lap-to-Table | 1, 8, 14 |
| Service experience | Friends and Family Test | 8 to 14 |
| System usability | System Usability Scale | 14 |

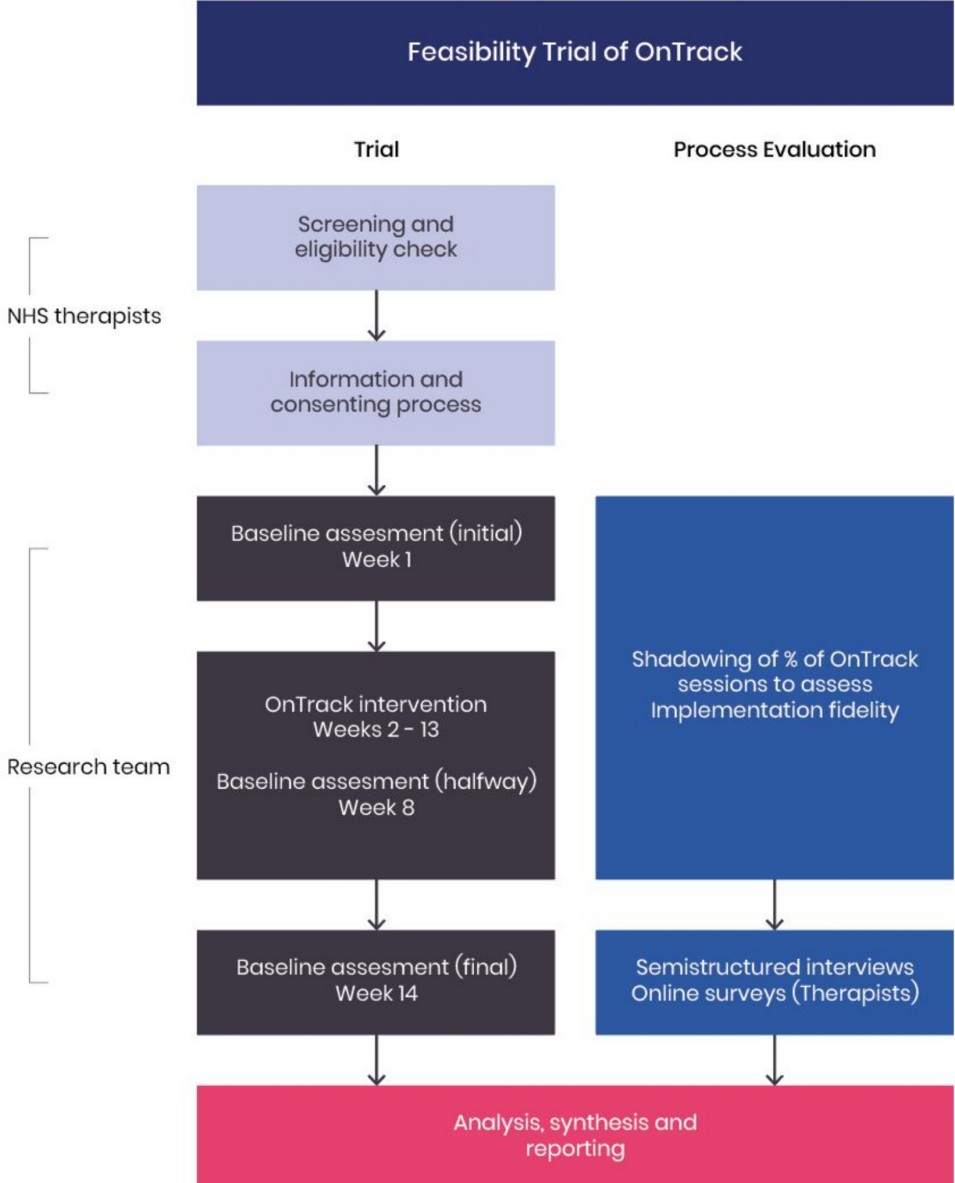

**Figure 1** Trial diagram. NHS, National Health Service.

## Study design

A feasibility study with a nested process evaluation (figure 1). The study is a single-site, non-randomised intervention trial. The design of the study was developed through a collaborative approach between the study researchers, a public and patient involvement (PPI) steering group, front-line therapists and the Research Design Service at the National Institute for Health Research.

An independent process evaluation will be conducted in parallel to learn about usage and engagement mechanisms of participants, therapists and other frontline staff, providing critical information for implementation fidelity and impact mechanisms necessary for scale-up.

## Study setting

The study will be conducted at an inner city NHS hospital Trust in London. Recruited participants will be able to continue to receive the intervention at home if discharged from hospital prior to ending the intervention period (14 weeks).

## Participants

The inclusion criteria encompasses:
► Adults (aged 18 or over).
► Stroke diagnosis less than 6 months previously (first or recurrent). Some participants will be recruited from an in-patient rehabilitation ward, hence the 6-month poststroke limit.
► Arm impairment of any type or level (including weakness—including dense hemiplegia, neglect and sensory deficits). This enables better understanding of which impairment level groups could benefit *or not* from using the intervention, especially considering the impact it may have on people's motivation regardless of their level of impairment.

- ► Ability to provide informed consent.
- ► Reliability to communicate (verbally or non-verbally) and understand English.
- ► Ability to read a predefined short message.

Potential participants who at the time of recruitment (or during participation) present with any of the following will be excluded:

- ► Unstable medical condition.
- ► Self-reported 'severe' pain in the arm affected either at rest or during movement.
- ► Severe oedema in the arm affected by their stroke, judged by the consenting therapist.
- ► Known discharge plans to a hospital other than the site Trust or residential care in less than 7 weeks (a small proportion of patients staying at Clinical Neurorehabilitation Unit (CNRU) may be in hospital for up to 12 weeks).
- ► Participants who are unable to engage with the intervention for a period of more than 7 consecutive days will be reviewed in a case-by-case basis by the members of the team responsible for delivering the intervention to determine if study continuation is appropriate.

## Recruitment

Participants will be recruited from the Hyperacute Stroke Unit (HASU), Acute Stroke Unit (ASU) and CNRU at an inner city NHS Hospital Trust in London.

Stroke therapists (occupational therapists, physiotherapists) will be responsible for screening and identifying suitable patients. They will introduce the study to potential participants and provide information documents. Potential participants will be given a minimum of 24 hours to consider the advantages and disadvantages of participating in the study and to formulate questions. Therapists will be able to answer questions or will liaise with the research team to provide an answer. Once all questions are answered and a potential participant is willing to participate, consent will be taken by the therapist. Only at this stage will patient information be shared with the research team. There may be situations where a therapist is only able to take verbal consent from a participant due to time or material constraints; in such cases, the researchers will be able to take written consent from the participant on first meeting them.

## Sample size calculation

Guidelines advocate a sample size of 12–30 participants for feasibility studies.[20] Experienced clinical academics and clinicians at the trial site have advised to expect about 50% of eligible patients to agree to participation and a 50% completion rate. This has informed a recruitment plan to identify at least 60 potential participants in a period of 30 weeks to reach the minimum sample size.

## Intervention

The intervention is the OnTrack system as a whole. The system consists of smart devices (smartphone and smartwatch), software (OnTrack app) and coaching support.

Smart devices are used to track arm movement. Motivational messages and a real-time display of completed arm activity (in minutes) are presented to the user via the OnTrack app. Coaching support is provided through fortnightly consultations by the researchers. During consultations, participants will receive self-management training informed by the Bridges Self-Management[21] and Taking Charge After Stroke[22] self-management programmes. Coaching sessions are themed around principles of self-management (see table 2, OnTrack consultation column).

Data gathered by the OnTrack system can be accessed by the researchers via a digital dashboard to inform consultations.

Participants will be loaned all equipment necessary for the trial and no previous experience with using smart devices is required to participate. Technical support will be provided only in cases where the hardware and/or software fail to perform the required functions to deliver the intervention.

Table 2 provides a participation schedule and a summary of the intervention procedures.

## Outcomes
### Feasibility of trial design and procedures

- ► Recruitment strategy and rates (feasibility of recruitment from HASU, ASU and CNRU wards)—percentage of patients: screened, eligible, approached, consented and excluded after screening. Participants consented and recruited will be logged by the research team in DOCUMAS.[23]
- ► Compliance and adherence to intervention—measure of minutes of activity per participant as recorded by the OnTrack app, engagement with OnTrack app as measured by system analytics (eg, compliance with starting tracking arm activity daily, number of times and times of the day a particular screen is visited, the number of messages read and replied to, etc).
- ► Completion rates—percentage of participants who complete the 14-week intervention period (not dropping out or being withdrawn from the study).
- ► Acceptability and reasons for decline/withdrawal— number of participants who withdraw or decline the intervention and reasons why. A record of reasons for withdrawal and declining will be kept by the researchers. Reasons will be categorised in order of most common; this information will help the research team to understand the reasons why someone might drop out or decline to participate in the study.

### Clinical assessments

As a secondary objective, clinical outcomes will be collected at different time points by a qualified member of the research team to identify an appropriate primary outcome, and to estimate parameters for a sample size calculation for an RCT (table 1). The outcome measures and assessments are listed next.

**Table 2** Intervention and participation schedule

| Week | Phase | Description | OnTrack consultation | Assessments |
|---|---|---|---|---|
| 0 | Information and consent | NHS therapists screen for eligible patients, provide information and consent participants | | Screening, information, and consent |
| 1 | Baseline assessment (initial) | Participants complete outcome measures and wear activity trackers (Axivity AX3) on both arms during waking hours (typically 12 hours/day) for 1 week to gather accelerometer data that are translated into minutes of activity. These data create a baseline of activity allowing left-to-right arm usage comparison | | PAM, FMA-UE, MAL, mRS, VAS, MoCA, AT, EQ-5D-5L, LTT |
| 2 | OnTrack intervention | Participants wear a smartwatch (Apple Watch Series 3 or 4) on their impaired arm only during waking hours (typically 12 hours/day). They will receive real-time feedback on the amount of movement completed (measured in minutes) and daily motivational messages. Participants will receive fortnightly consultations with a researcher to troubleshoot and receive self-management skills training. Baseline assessments are repeated during week 8 (halfway) | Onboarding | |
| 3 | | | Check-in and self-management skills training (problem solving) | |
| 4 | | | | |
| 5 | | | Check-in and self-management skills training (self-discovery) | |
| 6 | | | | |
| 7 | | | Check-in and self-management skills training (goal setting) | |
| 8 | | | Halfway assessment. Check-in and self-management skills training (goal setting cont.) | PAM, FMA-UE, MAL, mRS, VAS, MoCA, AT, EQ-5D-5L, LTT, FFT |
| 9 | | | | |
| 10 | | | Check-in and self-management skills training (reflection) | |
| 11 | | | | |
| 12 | | | Check-in and self-management skills training (sign-posting) | |
| 13 | | | | |
| 14 | Baseline assessment (exit) | Participants complete outcome measures and wear activity trackers (Axivity AX3) on both arms during waking hours (typically 12 hours/day) for 1 week to gather accelerometer data that are translated into minutes of activity. These data create a baseline of activity allowing left-to-right arm usage comparison | | PAM, FMA-UE, MAL, mRS, VAS, MoCA, AT, EQ-5D-5L, LTT, FFT, SUS |
| 15 | Feedback | Independent evaluator leads feedback sessions with participants who have completed the intervention. End of participation | | Semistructured interview, online survey (therapists) |

AT, Albert's Test; FMA-UE, Fugl-Meyer Assessment for Upper Extremity; FTT, Friends and Family Test; LTT, Lap-to-Table; MAL, Upper Extremity Motor Activity Log-14; MoCA, Montreal Cognitive Assessment; mRS, modified Rankin Scale; NHS, National Health Service; PAM, Patient Activation Measure; SUS, System Usability Scale; VAS, Visual Analogue Scale.

## Patient activation

Patient activation is a concept recognised by the NHS that describes the knowledge, skills and confidence a person has in managing their own health and healthcare.[24] This will be measured using the Patient Activation Measure (PAM)[25] which has been validated in stroke populations in the UK.[26] The PAM survey measures patients on a 0–100 scale and can categorise patients into one of four activation levels along an empirically derived continuum.[25] Activation levels will be used to allocate participants one of three different OnTrack coaching tiers. The tiers aim to make the different aspects of the coaching more relevant and meaningful for the individual participant and their stage of recovery and self-management.

## Arm impairment

Arm impairment will be measured objectively using the Fugl-Meyer Assessment for Upper Extremity (FMA-UE).[27] The FMA-UE has been tested extensively and is found to have excellent psychometric properties and is recommended as core measures to be used in every stroke recovery and rehabilitation trial.[28]

## Arm function

Arm function will be assessed using the Upper Extremity Motor Activity Log-14 (MAL).[29] The MAL is a scripted, structured interview developed to self-report the amount and quality of use of the impaired arm in individuals with stroke in 14 different activities of daily living.

## Gross level of disability

The modified Rankin Scale[30] is the most prevalent functional outcome measure in contemporary stroke trials. The mRS quantifies disability using an ordinal hierarchical grading from 0 (no symptoms) to 5 (severe disability).

## Arm pain

Pain will be assessed using a Visual Analogue Scale (VAS) from 0 (no pain) to 10 (excruciating pain) over the last 24 hours. VAS is a valid measure of pain intensity and is responsive to change.[31] Individuals scoring 3/10 or more in the affected arm will be withdrawn from the study unless their pain is only on movements that are not part of their usual everyday activities (eg, arm pain when doing overhead reaching).

## Cognitive impairment

Cognitive impairment will be assessed using the Montreal Cognitive Assessment (MoCA). The MoCA is a brief cognitive screening tool with high sensitivity and specificity for detecting mild cognitive impairment.[32] The MoCA defines impairment as follows: score of 18–25=mild, 10–17=moderate and <10 = severe.[32] Participants' scores will be used to look for associations between the use of OnTrack and any cognitive impairment.

## Perceptual neglect

Albert's Test is being used to assess for unilateral spatial neglect (USN). This a simple test where participants are asked to cross out lines ruled in a standard fashion on a sheet of paper. If any lines are left uncrossed, and more than 70% of uncrossed lines are on the same side as motor deficit, USN is indicated. This may be quantified in terms of the percentage of lines left uncrossed. The test is very easy to administer and is a good predictor of functional activity 6 months after stroke onset.[33]

## Quality of life

The EQ-5D-5L is a widely used standardised preference-based measure of health status developed by the EuroQol Group in order to provide a simple, generic measure of health for clinical and economic appraisal.[34]

## Additional assessments

A Lap-to-Table timed test will be performed where the researchers measure the time it takes a participant to move their hand three times from resting on their lap to a table positioned in front of them. This test is performed to assess its potential to use as part of the inclusion criteria for an RCT.

The NHS Friends and Family Test[35] will be used to obtain feedback on the overall experience of using OnTrack and participating in the trial. Participants will be asked: 'How likely are you to recommend OnTrack to friends and family if they needed similar care or treatment?' with answers provided in a Likert 5-point scale ranging from 'extremely likely' to 'extremely unlikely' and an 'I don't know' option.

The System Usability Scale (SUS) will be used to subjectively assess the usability of the OnTrack intervention. The test is a simple, 10-item scale covering a variety of aspects of system usability, such as the need for support, training and complexity, and thus have a high level of face validity for measuring the usability of a system.[36]

## Process evaluation

A process evaluation will be carried out by researchers working independently to the intervention team and in parallel to the trial to determine whether the OnTrack intervention was delivered as intended and to understand the mechanisms of impact. The aim of the process evaluation at the feasibility stage is mainly to understand how the trial design and intervention could be optimised ahead of an RCT.[37] A logic model[38 39] that defines the intervention in terms of inputs, outputs, causal assumptions and expected outcomes has been developed to help identify core questions for the evaluation team to explore (figure 2). The evaluation team will observe 10% of all intervention sessions with the objective of documenting fidelity, dose and reach of the intervention.

Critical reflection and the process evaluation will help refine the intervention, as shown by mid-range theories (ie, theories that help understand implementation).[40] Interim results will be shared with the intervention team at the half-way point with the objective to review some of the procedures and make minor adjustments as necessary.

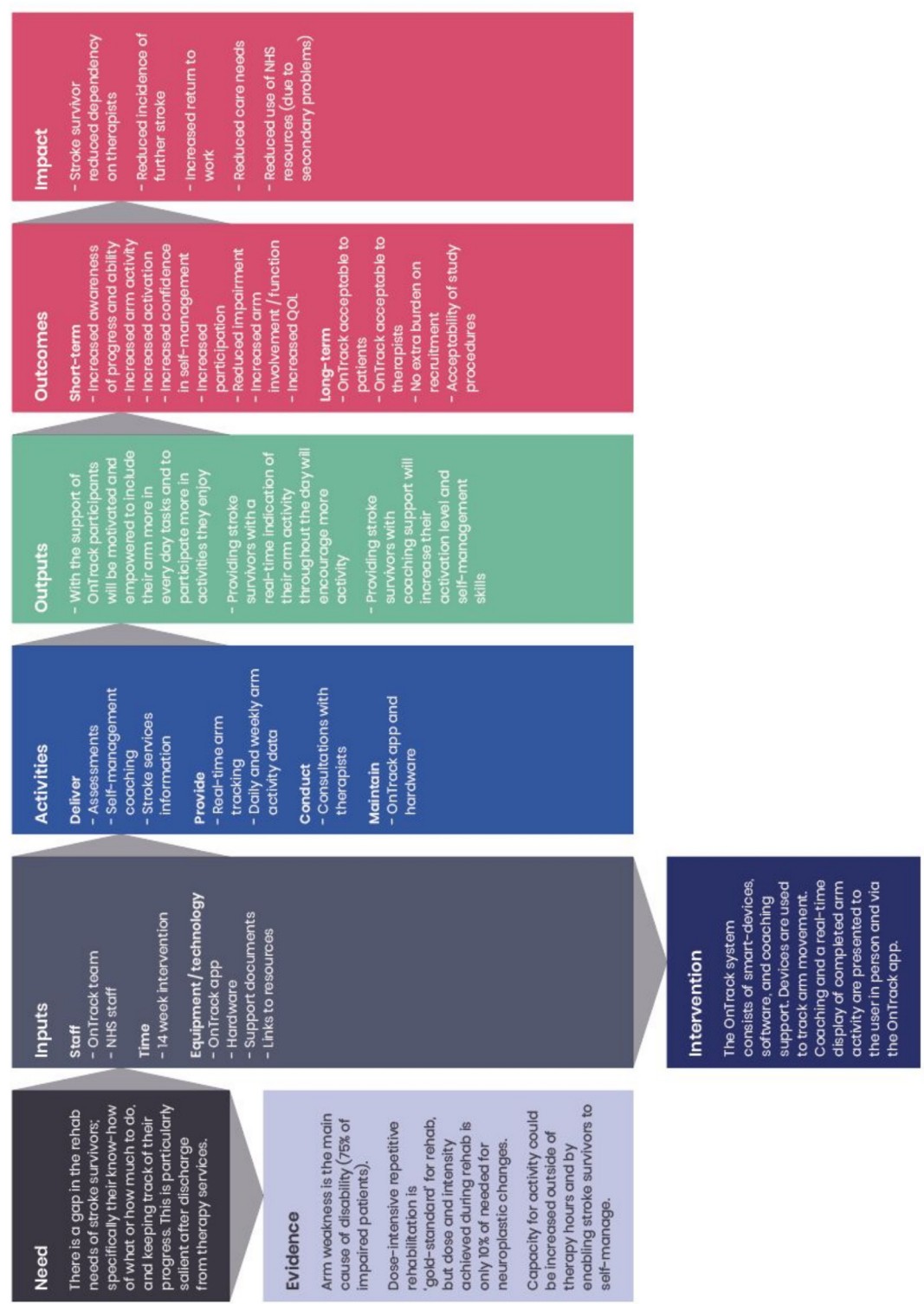

**Figure 2** Logic model.

In-depth semistructured interviews will be conducted with patients at the end of their participation; a minimum sample of 12 is anticipated. A topic guide with themes drawing from the logic model will be used. Interviews will focus on participants' experiences using OnTrack, their perceptions of arm tracking, motivational messaging and the researcher consultations. Additionally, the interviews will explore participants' perceptions of the impact OnTrack had on them in terms of progress, awareness, participation and confidence in self-management. Participants' responses will be compared against activity data collected from the OnTrack app.

NHS therapists caring for participants taking part will be consented and invited to complete a short online survey to gather their feedback regarding acceptability of study procedures and they have the option to respond

anonymously. The total number of therapists involved is difficult to predict as there may be team changes and staff movement during the course of the study. The survey will ask questions around three themes: (1) participation, relevance, quality and time spent in study procedures; (2) opinions on the benefit/detriment OnTrack may have for patients and (3) opinions on how the intervention may or may not fit with service provision and their workflow.

### Data analysis

Analysis will be completed on the parameters and implementation of the study in addition to the usability of OnTrack.

Data collected for the process evaluation will be a combination of qualitative data from interviews with participants who had a stroke and therapists to explore their experiences of using OnTrack, as well as quantitative data on the usage of OnTrack and the self-reported SUS. OnTrack therapy support sessions will be monitored through a fidelity checklist and observations (10 live sessions will be observed in total). In addition, the evaluation team will have access to recorded sessions that can be observed at their discretion). Interview data will undergo thematic analysis by the evaluation team. Data will be entered into NVIVO[41]; line-by-line coding and analysis will be informed by Braun and Clark's approach to thematic analysis.[42]

Changes over time will be evaluated in both OnTrack usage and outcome measures.

For OnTrack usage, the team will analyse users' activity patterns by day and hour of day. Figure 3 illustrates examples of visualisations created using aggregated data captured by OnTrack from healthy beta testers between June and August 2019. It compares users on active minutes per hour of day (aggregated over time) and active minutes per day.

OnTrack also captures specific usage metrics, including:

▶ Number of times OnTrack messages were opened.
▶ Number of times daily and weekly activity were viewed on the phone.
▶ Number of swipes on watch to reveal activity graph.

For each patient, we will plot the values above against their minutes of activity to better understand the potential impact of the app on activity over time.

The self-reported PAM will be captured at weeks 1, 8 and 14 for each user. It will be analysed in relation to the minutes of activity of each user over time to better understand the potential impact of the app on their levels of activation. SUS will be captured at weeks 8 and 14 and will be compared against actual usage metrics (described above) to assess usability.

While conducting meaningful significant subgroup analyses would be difficult given the relatively small sample size, we believe that outputs from this study could potentially inform the subgroups that might be considered for inclusion in a larger trial.

All data will be stored and accessed in accordance with General Data Protection Regulation (GDPR) guidance.

Clinical trial support will be provided by the Big Data and Analytical Unit at Imperial College London's Institute of Global Health Innovation.

## PATIENT AND PUBLIC INVOLVEMENT

To date, over 100 stroke survivors, carers and therapists have been involved in the design of OnTrack. Participants have been instrumental in highlighting areas for improvement in upper limb stroke rehabilitation. They have contributed to a codesign process (including workshops, interviews, observations and surveys) resulting in the design, development and initial testing of OnTrack.

A steering group comprising four stroke survivors was formed for the purpose of this feasibility study. Diversity within the group—both in terms of demographics and stroke severity—was considered. The group has supervised the development of all patient-facing material ensuring its clarity. They will also participate in data analysis by helping to refine themes and key messages arising from qualitative interviews. Participants will be trained by experienced researchers for this purpose.

The steering group will meet five times over the duration of the study, including an initial briefing session at the start to outline their involvement. Steering group members will be key members of the research team and their time and travel will be reimbursed according to the National Institute for Health Research INVOLVE[43] guidelines.

The PPI involvement plan was shared with Imperial College London's PPI 'Research Partners Group' on 21 February 2019 who felt that the needs of the steering group have been accounted for.

## ETHICS AND DISSEMINATION

The OnTrack study will be conducted in accordance with the recommendations for physicians involved in research on human subjects adopted by the 18th World Medical Assembly, Helsinki 1964 and later revisions; and in compliance with the relevant UK and European legislation including the NHS Health Research Authority (HRA) policy frameworks and the GDPR 2018.

The study was granted ethical approval by the HRA, Health and Care Research Wales, and the London–Surrey Research Ethics Committee (ref. 19/LO/0881). Local site capacity and capability approval has been granted by the hospital Trust.

The current approved protocol version is V.1.3 dated 19 June 2019. Protocol amendments will be submitted for approval to the NHS HRA in the first instance and to the local site thereafter ahead of implementation.

The chief investigator is responsible for preserving the confidentiality of participants taking part in the study. Researchers will have patients' names, contact numbers, emails and home addresses for the purposes of arranging visits. This information will be stored in accordance with GDPR legislation. Participants are free to withdraw from

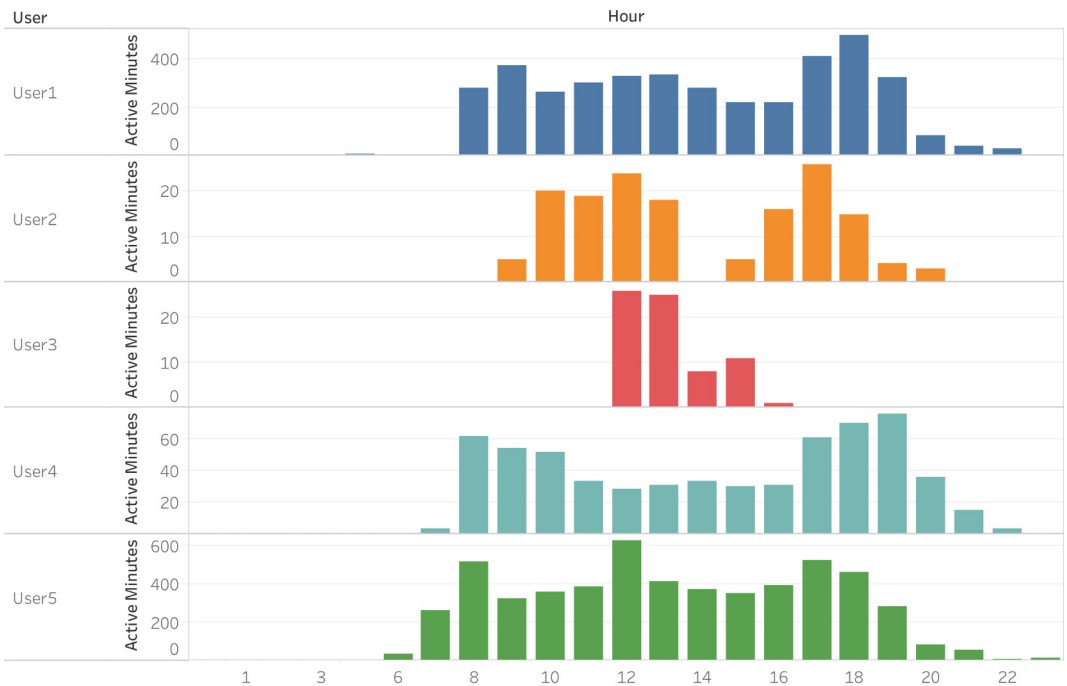

Active minutes per hour of day (0-23) by user, aggregated over time

Source: Sample OnTrack usage data collected from healty beta testers, June-August 2019

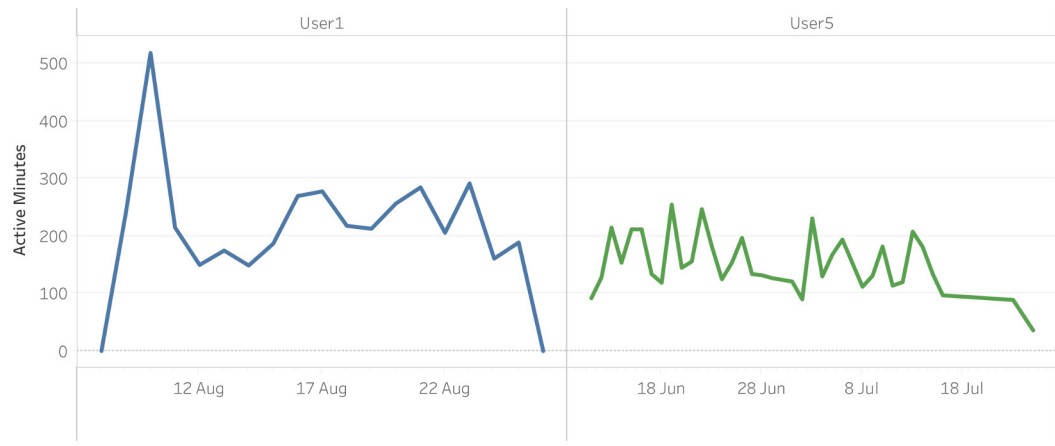

Active minutes per day by user

Source: Sample OnTrack usage data collected from healthy beta testers, June-August 2019

**Figure 3** Examples of visualisations created using aggregated data captured by OnTrack from healthy beta testers. Data for a minimum of 5 and a maximum of 18 days were aggregated for the period between June and August 2019.

the study at any time. However, anonymised activity data collected may still be used for data analysis as this is unlinked of any patient identifiable information.

The day-to-day management of the study will be coordinated by the Helix Centre. A study steering committee formed by the intervention team, evaluation team, PPI group and representatives from the local site will meet at regular intervals throughout the study.

Regular updates about the trial will be made available through social media, blog posts, newsletters and the Helix Centre website (www.helixcentre.com). Trial results will be submitted for publication in journals, presented at national and international stroke meetings and conferences and disseminated among stroke communities.

**Trial status**

The first participant was enrolled on 9 September 2019 and recruitment is expected to complete by the end of March 2020. Enrolment and data collection were continuing as planned at the time of submission of this protocol.

**Acknowledgements** The authors would like to acknowledge the contributions of the members of the PPI group; Jennifer Crow and Sarah Daniels for their input on recruitment and sample size calculations; and Gaby Judah for her help in defining some aspects of the intervention.

**Contributors** AD is grant holder and has project oversight along with DD. GF, EG and LH developed the intervention and conceived of the study. GF, EG, and FJ initiated the study design and ET and ML helped with further refinement. EG and GF are responsible for delivering the intervention and data collection. FJ and ET are responsible for the process evaluation. ML provides statistical expertise in trial design and is conducting the primary statistical analysis. All authors contributed to the refinement of the study protocol and approved the final manuscript.

**Funding** This study is funded by the NIHR Imperial Biomedical Research Centre, grant 1215-20013. All intellectual property associated with the OnTrack system is owned by the Helix Centre which is a collaboration between Imperial College London and the Royal College of Art.

**Disclaimer** The views expressed are those of the author(s) and not necessarily those of the NIHR or the Department of Health and Social Care.

**Competing interests** FJ is the founder of the social enterprise Bridges Self-Management. She has not received any financial support for this work that could have influenced the design.

**Patient and public involvement** Patients and/or the public were involved in the design, or conduct, or reporting, or dissemination plans of this research. Refer to the Methods section for further details.

**Patient consent for publication** Not required.

**Provenance and peer review** Not commissioned; externally peer reviewed.

**ORCID iDs**
Gianpaolo Fusari http://orcid.org/0000-0002-7263-3398
Elizabeth Taylor http://orcid.org/0000-0002-4596-823X

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
