## [Reviewer comments · BMJ Open]

ARTICLE DETAILS

TITLE (PROVISIONAL)	Protocol for a feasibility study of OnTrack: a digital system for upper-limb rehabilitation after stroke
AUTHORS	Fusari, Gianpaolo; Gibbs, Ella; Hoskin, Lily; Dickens, Daniel; Leis, Melanie; Taylor, Elizabeth; Jones, Fiona; Darzi, Ara

VERSION 1 – REVIEW

REVIEWER	Iris Brunner Aarhus University, Hammel Neurocenter, Denmark
REVIEW RETURNED	23-Nov-2019

GENERAL COMMENTS	Very thoroughly planned feasibility study. I especially appreciate the process evaluation. Some remarks to details, which are not clear to me: Exclusion criteria: "Known discharge plans to a hospital other than the site Trust or residential care in less than 7 weeks. " Do patients really stay that long, frequently more than 7 weeks? Do I misunderstand anything? Could the authors make that clearer? Patients with any type of arm weakness are included. Do the authors really expect patients with paralysis to move their arm? The intervention period is 14 weeks. For how long time do the patients have to wear the watch each day, during waking hours? Please specify. Could the authors give some examples for the self-management training?
--

REVIEWER	Cathy Stinear University of Auckland, New Zealand
REVIEW RETURNED	27-Nov-2019

GENERAL COMMENTS	Introduction, 2nd paragraph: "A recent Cochrane review of over 500 trials failed to yield high-quality practice recommendations.(6)" Reference 6 is 15 years old, and therefore not recent. It included 151 studies, not "over 500". Please be more accurate in your description, or select another study to cite in support of this statement. Please clarify whether the "practice recommendations" are specifically for the upper limb. Introduction, 2nd paragraph: "Arm recovery after stroke is a national research priority,(7) nonetheless, studies suggest that the actual time patients spend exercising is minimal.(8,9)" References 8 and 9 appear to relate to overall physical activity, rather than activity involving the upper limb. So how does the first part of this sentence relate to the second? Introduction, 3rd paragraph: "An ethnographic study conducted by
---

	the Helix Centre(16) (funded by Innovate UK) revealed that patients struggle to see and keep track of improvements, this impacts their motivation and leaves them dependent on therapists for feedback." Reference 16 is a website that provides no information to support the statements made here. The term "repetitive rehabilitation" is used in several places, but its meaning is unclear. Introduction, last paragraph: "A 20% mean increase in activity was observed." Is this upper limb activity? What are the units of measurement? Please cite a reference in support of this assertion, so the reader can evaluate the data that provide the rationale for the present study. The next sentence talks about using the OnTrack System to "increase the overall amount of arm rehabilitation received". However, it seems the system is designed to increase the use of the affected upper limb in everyday activities, rather than increasing the amount of rehabilitation "received" by patients from therapists. In light of the comments above, the rationale is not robust, and deserves a more rigorous explanation. Methods, Participants: If testing feasibility of recruitment from the hyperacute and acute settings is one of the aims, however the inclusion criteria state that participants need to be less than six months post-stroke. This is slightly confusing, and a rationale could be provided to help the reader. Methods, Participants: How are "severe" pain and oedema defined? Later, a report of pain greater than 3/10 is defined as an exclusion criterion. This doesn't seem to be "severe", and ought to be listed as an exclusion criterion. Methods, Sample Size Calculation: Please define the minimum sample size here. The abstract notes that the target is N = 24. However, if 60 potential participants are expected to be identified in 30 weeks, and half of these are expected to agree to take part (N = 30) and half of these are expected to complete, this will mean only 15 people completing the intervention. How does this relate to the sample size of 24 provided in the Abstract? Methods: The information provided about the tracking of arm movements, the parameters derived from these measures, the real-time display, motivational messages, and the coaching support, is not sufficiently detailed to enable replication. Methods, Outcome Measures, Feasibility: Who will keep a record of screened patients? How is "engagement" with the OnTrack system measured? There isn't any information provided about the OnTrack system, how it collects data, how these data are processed, or what measures are derived. How is "completion" of the intervention defined? How will "reasons why" for withdrawal or declining the intervention be analysed and reported? Methods, Clinical Assessments: Who will carry out these assessments? What are the three "coaching tiers"? What MoCA score will be used to define cognitive impairment? Similarly, what is the threshold for identifying perceptual neglect?
--	--

	Methods: Please provide information about the survey to be completed by NHS therapists. Will it be anonymous? Will they provide consent as participants in this research? How many therapists will be involved? What sorts of questions will the survey ask? Data Analysis, second paragraph: "changes over time" in what? Please provide more information regarding the planned subgroup analyses. Which patient demographics? How is "stroke disability" measured and how will it be used for subgroup analysis? With a sample of 24 (though this is unclear) how many subgroups will be available for analyses? Does the measurement of upper limb activity (in minutes) distinguish between upper limb use and arm swing during gait? Please provide more information about the use of the bilateral activity trackers used for one week. What data will be collected? How will these data be analysed? What percentage of OnTrack sessions will be observed to evaluate fidelity? Figure 3: Over how many days were these data aggregated for each participant? The sponsor is unclear. Is there any intellectual property associated with the OnTrack System?
--	--

VERSION 1 – AUTHOR RESPONSE

Reviewer 1

Reviewer Name: Iris Brunner

Institution and Country: Aarhus University, Hammel Neurocenter, Denmark

Please state any competing interests or state 'None declared': None declared

Exclusion criteria: "Known discharge plans to a hospital other than the site Trust or residential care in less than 7 weeks. " Do patients really stay that long, frequently more than 7 weeks? Do I misunderstand anything? Could the authors make that clearer?

Participants section has been amended.

Patients with any type of arm weakness are included. Do the authors really expect patients with paralysis to move their arm?

Participants section has been amended.

The intervention period is 14 weeks. For how long time do the patients have to wear the watch each day, during waking hours? Please specify.

Table 2 has been amended to reflect this.

Could the authors give some examples for the self-management training?

The text in the Intervention section has been amended.

Reviewer 2

Reviewer Name: Cathy Stinear

Institution and Country: University of Auckland, New Zealand

Please state any competing interests or state 'None declared': None declared

Introduction, 2nd paragraph: "A recent Cochrane review of over 500 trials failed to yield high-quality

practice recommendations.(6)" Reference 6 is 15 years old, and therefore not recent. It included 151 studies, not "over 500". Please be more accurate in your description, or select another study to cite in support of this statement. Please clarify whether the "practice recommendations" are specifically for the upper limb.

Noted, there has been an error in the way the citations were listed. The Cochrane review referred to here should correlate to reference 4. All affected citation numbers have been corrected.

With regards to the practice recommendations, a more specific description has been added to the text.

Introduction, 2nd paragraph: "Arm recovery after stroke is a national research priority,(7) nonetheless, studies suggest that the actual time patients spend exercising is minimal.(8,9)" References 8 and 9 appear to relate to overall physical activity, rather than activity involving the upper limb. So how does the first part of this sentence relate to the second?

Text has been amended

Introduction, 3rd paragraph: "An ethnographic study conducted by the Helix Centre(16) (funded by Innovate UK) revealed that patients struggle to see and keep track of improvements, this impacts their motivation and leaves them dependent on therapists for feedback." Reference 16 is a website that provides no information to support the statements made here.

Text has been amended

The term "repetitive rehabilitation" is used in several places, but its meaning is unclear.

A definition has been added to the text

Introduction, last paragraph: "A 20% mean increase in activity was observed." Is this upper limb activity? What are the units of measurement? Please cite a reference in support of this assertion, so the reader can evaluate the data that provide the rationale for the present study. The next sentence talks about using the OnTrack System to "increase the overall amount of arm rehabilitation received". However, it seems the system is designed to increase the use of the affected upper limb in everyday activities, rather than increasing the amount of rehabilitation "received" by patients from therapists. The text has been amended. Please note that the results of these tests have not been published yet therefore an academic citation is unavailable.

In light of the comments above, the rationale is not robust, and deserves a more rigorous explanation.

Methods, Participants: If testing feasibility of recruitment from the hyperacute and acute settings is one of the aims, however the inclusion criteria state that participants need to be less than six months post-stroke. This is slightly confusing, and a rationale could be provided to help the reader.

We are also recruiting from a local Clinical Neurorehabilitation Unit (CNRU), patients in this ward may come from via different referral routes and NHS settings and not necessarily from an acute ward, hence the 6-month time limit.

Methods, Participants: How are "severe" pain and oedema defined? Later, a report of pain greater than 3/10 is defined as an exclusion criterion. This doesn't seem to be "severe", and ought to be listed as an exclusion criterion.

Pain: the text has been updated to reflect an exclusion criteria for patients who self report "severe" pain in the arm at rest or during movement during screening. Screening therapists will not be performing the VAS pain scale and therefore the score of 3/10 or higher is not listed as an exclusion criterion. Once the researchers assess arm pain using the VAS scale, then any participant scoring 3/10 or higher will be withdrawn from the study. The text has been updated, the word "excluded" has been removed.

Oedema: this will be based on the consenting therapist's judgement of the individual patient. The text reflects this now.

Methods, Sample Size Calculation: Please define the minimum sample size here. The abstract notes that the target is N = 24. However, if 60 potential participants are expected to be identified in 30 weeks, and half of these are expected to agree to take part (N = 30) and half of these are expected to complete, this will mean only 15 people completing the intervention. How does this relate to the sample size of 24 provided in the Abstract?

The minimum sample size advised by the literature for feasibility studies is 12 (Billingham et al), the

abstract has been amended.

Methods: The information provided about the tracking of arm movements, the parameters derived from these measures, the real-time display, motivational messages, and the coaching support, is not sufficiently detailed to enable replication.

The OnTrack system is used for the tracking of arm movement, real-time display of minutes of activity and motivational messaging. Coaching support is provided by the OnTrack team (the research team in this study). The OnTrack system is needed in order to replicate the study.

Methods, Outcome Measures, Feasibility: Who will keep a record of screened patients? How is "engagement" with the OnTrack system measured? There isn't any information provided about the OnTrack system, how it collects data, how these data are processed, or what measures are derived. How is "completion" of the intervention defined? How will "reasons why" for withdrawal or declining the intervention be analysed and reported?

The text has been amended

Methods, Clinical Assessments: Who will carry out these assessments? What are the three "coaching tiers"? What MoCA score will be used to define cognitive impairment? Similarly, what is the threshold for identifying perceptual neglect?

The text has been amended

Methods: Please provide information about the survey to be completed by NHS therapists. Will it be anonymous? Will they provide consent as participants in this research? How many therapists will be involved? What sorts of questions will the survey ask?

The text has been amended

Data Analysis, second paragraph: "changes over time" in what? Please provide more information regarding the planned subgroup analyses. Which patient demographics? How is "stroke disability" measured and how will it be used for subgroup analysis? With a sample of 24 (though this is unclear) how many subgroups will be available for analyses?

The text has been amended

Does the measurement of upper limb activity (in minutes) distinguish between upper limb use and arm swing during gait?

The proprietary algorithm that we have developed using movement data from stroke survivors is capable of distinguishing between arm swing during gait and other upper limb activity. Arm motion during walking and transport does not account for minutes of activity measured using OnTrack.

Please provide more information about the use of the bilateral activity trackers used for one week. What data will be collected? How will these data be analysed?

The text has been amended

What percentage of OnTrack sessions will be observed to evaluate fidelity?

The text has been amended

Figure 3: Over how many days were these data aggregated for each participant?

The text has been amended

The sponsor is unclear. Is there any intellectual property associated with the OnTrack System?

The text has been amended

VERSION 2 – REVIEW

REVIEWER	Iris Brunner Aarhus University, Hammel Neurocenter
REVIEW RETURNED	11-Jan-2020
GENERAL COMMENTS	I still think the planned study is very interesting. The in-depth description of the planned assessment and process evaluation is instructive for other researchers. The concerns have been addressed. Just a few remarks:

	- The targeted number of a minimum of 12 are rather few participants. The authors earlier stated to include 24 patients. It can be discussed if publishing a protocol of such a small study makes sense. Nevertheless, the thorough description of the mixed methods applied is valuable and worth publishing in my opinion. - Data analysis: The authors mention that subgroup analysis are planned. This is not meaningful with such a small sample. - Figure 3 is missing in the revised version.
--	---

REVIEWER	Cathy Stinear University of Auckland, New Zealand
REVIEW RETURNED	05-Feb-2020

GENERAL COMMENTS	Introduction, 2nd paragraph. The authors have amended the text, however the ideas in this paragraph remain unclear. The text is now as follows: “Arm recovery after stroke is a national research priority.(6) There is a correlation between physical activity after stroke and the ability to perform activities of daily living (most of which involve the use of the arm) nonetheless, studies suggest that the actual time patients are active is minimal.(7,8) Many current approaches to solving this problem focus on improving the prescribed rehabilitation sessions, often employing gamification techniques.(9,10)” What point is being made when the authors identify that there’s a correlation between physical activity and the ability to perform ADLs (which is unsurprising), “nonetheless” the actual time patients are active is minimal? And what is “this problem” referred to in the following sentence? Is the problem one of low physical activity levels, reduced ADL abilities, or both, or the relationship between them? Please be clear about whether general physical activity or real-world use of the upper limb is being discussed here, and how this relates to the study’s rationale. Introduction, 3rd paragraph. Thank you for clarifying that the in-house ethnographic study has not been published. Please make this clear as follows: “An unpublished ethnographic study conducted by the Helix Centre..” The inclusion criteria have expanded on the rationale for including people with arm impairment of any type or level (the number of expected “levels” of impairment is not identified). However, it’s hard to see how any meaningful insights will be gained from a small sample size (as few as 12), as there will probably only be one or two people with each type or level of arm impairment. Subgroup analyses are also planned based on the MoCA score, patient demographics, mRS score, stroke subtype, and the care pathway. The authors acknowledge that the number of subgroups available for analysis will be small (and each subgroup will have very few patients). However it’s hard to see how a sample of as few as 12 heterogeneous patients will meaningfully inform inclusion/exclusion criteria or the analysis plan for a future RCT. The authors expect around 15 people to take part. This might be a large enough sample to gain some understanding of the barriers and facilitators to recruitment and retention in a future trial. But it’s probably not large enough to determine the specific demographic or clinical characteristics that could be used for future inclusion/exclusion criteria or subgroup analyses. The method for combining participants’ data for analysis is not
--

	provided, either for the whole group or sub-groups. Please operationally define “completion” of the 14-week intervention period. Does this just mean not dropping out before 14 weeks? Or something else? The OnTrack coaching tiers have not been explained – what are they? How will participant experience differ between tiers? Can a participant’s coaching tier change during the study?
--	---

VERSION 2 – AUTHOR RESPONSE

Reviewer: 1

Reviewer Name: Iris Brunner

Institution and Country: Aarhus University, Hammel Neurocenter, Denmark

Please state any competing interests or state ‘None declared’: None declared

I still think the planned study is very interesting. The in-depth description of the planned assessment and process evaluation is instructive for other researchers. The concerns have been addressed.

Just a few remarks:

The targeted number of a minimum of 12 are rather few participants. The authors earlier stated to include 24 patients. It can be discussed if publishing a protocol of such a small study makes sense. Nevertheless, the thorough description of the mixed methods applied is valuable and worth publishing in my opinion.

In the original manuscript we stated that we would aim to recruit a minimum of 24 participants with the expectation of having a 50% drop out rate. This would result in the minimum sample size of 12 that we are expecting and that is advised by the NIHR. We acknowledge that the original wording was potentially confusing, hence the change in the text.

Data analysis: The authors mention that subgroup analysis are planned. This is not meaningful with such a small sample.

This section has been revised, the subgroup analysis has been removed as the sample size is too small to do a meaningful analysis. However, this study can potentially inform the subgroups that we might consider for a larger trial.

Figure 3 is missing in the revised version.

Noted and corrected.

Reviewer: 2

Reviewer Name: Cathy Stinear

Institution and Country: University of Auckland, New Zealand

Please state any competing interests or state ‘None declared’: None declared

Introduction, 2nd paragraph. The authors have amended the text, however the ideas in this paragraph remain unclear. The text is now as follows: “Arm recovery after stroke is a national research priority.(6) There is a correlation between physical activity after stroke and the ability to perform activities of daily living (most of which involve the use of the arm) nonetheless, studies suggest that the actual time patients are active is minimal.(7,8) Many current approaches to solving this problem focus on improving the prescribed rehabilitation sessions, often employing gamification techniques.(9,10)” What point is being made when the authors identify that there’s a correlation between physical activity and the ability to perform ADLs (which is unsurprising), “nonetheless” the actual time patients are active is minimal? And what is “this problem” referred to in the following sentence? Is the problem one of low physical activity levels, reduced ADL abilities, or both, or the

relationship between them? Please be clear about whether general physical activity or real-world use of the upper limb is being discussed here, and how this relates to the study's rationale.

The point we are attempting to make in the manuscript is that despite knowing that more activity could translate to better arm function, patients still remain largely inactive (especially when not "doing rehab" - a subject we touch upon further down the text). Moreover, therapy in hospital often prioritises mobility over arm recovery even when arm recovery has been identified by stroke survivors, carers and clinicians to be a national research priority.

The intervention aims to increase activity in the upper limb, therefore we believe that the relationship between inactivity and ability to perform ADLs is relevant. If the current study is successful, our intent is to investigate through an RCT whether more general activity in the arm (as measured via the OnTrack app) will correlate to improved function (as measured by the Fugl-Meyer) and more and better ADL performance (as measured through the MAL).

The text has been amended.

Introduction, 3rd paragraph. Thank you for clarifying that the in-house ethnographic study has not been published. Please make this clear as follows: "An unpublished ethnographic study conducted by the Helix Centre.."

The text has been amended

The inclusion criteria have expanded on the rationale for including people with arm impairment of any type or level (the number of expected "levels" of impairment is not identified). However, it's hard to see how any meaningful insights will be gained from a small sample size (as few as 12), as there will probably only be one or two people with each type or level of arm impairment. Subgroup analyses are also planned based on the MoCA score, patient demographics, mRS score, stroke subtype, and the care pathway. The authors acknowledge that the number of subgroups available for analysis will be small (and each subgroup will have very few patients). However it's hard to see how a sample of as few as 12 heterogeneous patients will meaningfully inform inclusion/exclusion criteria or the analysis plan for a future RCT. The authors expect around 15 people to take part. This might be a large enough sample to gain some understanding of the barriers and facilitators to recruitment and retention in a future trial. But it's probably not large enough to determine the specific demographic or clinical characteristics that could be used for future inclusion/exclusion criteria or subgroup analyses. We agree with this assessment. The present study will help the researchers narrow down an idea of who might benefit most from the intervention as well as who might not benefit much from it. Whereas the number of participants may not be large enough to generate insights from quantitative data, an awful lot can be learned from qualitative methods and the process evaluation that is being conducted independently. We recognise that conducting any significant subgroup analysis will be difficult given the small sample size, nevertheless, the outputs from this study can potentially inform the subgroups that we might consider for a larger trial.

The text has been amended

The method for combining participants' data for analysis is not provided, either for the whole group or sub-groups.

We will not be combining participants' data for analysis. We will be looking at each participant's minutes of activity against their usage data and outcome measures.

The text under Data Analysis has been amended to reflect this.

Please operationally define "completion" of the 14-week intervention period. Does this just mean not dropping out before 14 weeks? Or something else?

Correct, this means not dropping out before 14 weeks or not being withdrawn from the study due to force majeure causes that would prevent the participant from engaging with the intervention for a period of more than 7 consecutive days (in such cases, the research team and participant will discuss if appropriate to continue).

The text has been amended

The OnTrack coaching tiers have not been explained – what are they? How will participant experience differ between tiers? Can a participant's coaching tier change during the study?

The coaching tier is derived from the PAM (which is taken at week 1, week 8 and week 14), a

participant may change tier if their results change at week 8. The differences between tiers are subtle but important; they mainly dictate the way the researchers approach the individual and the content/frequency of the digital messages on the app.

For example, a participant with lower activation may receive a message 7 days per week whilst a higher activation participant may only receive the same type of message 5 days per week (under the hypothesis that a lower activation participant will require more encouragement). Similarly, face to face sessions with a lower activation participant may focus on one simple task to achieve during the week whilst a higher activation participant may be able to manage two or more tasks at the same time. We believe that the overall experience of the intervention would not be significantly different for the individual as the overarching aim is to increase activity in the arm regardless of the tier.